# Examining Collaborative Fanfiction: New Practices in Hyperdiegesis and Poaching

## Abby Kirby

Independent Researcher, Kenosha, WI 53140, USA; kirbyabby7@gmail.com

**Abstract:** This paper focuses on how collaborative fanfiction has taken on new practices to accommodate fans as they gather new spaces for online communication as well as desire a deeper sense of community. Collaborative subcultures involve large groups of fans who work together to create expansive world-building for their fanfictions, or even create new fandoms from scratch. In order to accommodate the vast amounts of ideas and stories that enter their communities, they have adapted hyperdiegetic narratives in order to write stories that are "believable" for a concept rather than adhere to a rigid canon. They also develop a culture of inter-fan poaching, which allows them to borrow an idea from another fan for their own stories, without the need for permission.

**Keywords:** fanfiction; hyperdiegesis; fandom; inter-fan poaching





## 1. Introduction

Fanfiction is often considered an individualistic enterprise. That is to say that when fans write fanfiction, they tend to emphasize their own desires and wishes rather than what others want. Fans often write by themselves and for themselves. In *Textual Poachers* (Jenkins 1992), Henry Jenkins writes that "fans tend to see themselves in highly individualistic terms, emphasizing their refusal to conform to 'mundane' social norms and the range of different interpretations circulating within their community" (p. 88).

While there have always been opportunities for fans to collaborate in fanfiction: fans who write a single fanfiction with close friends, write as a gift for another person, or take on a beta reader, it is a rare occurrence to see fans collaborating and connecting their fanfiction to another. However, in this article, I refer to collaborative fanfiction as a mass production, in which large groups of fans all create individual stories, chapters, or segments revolving around one concept. This collaborative fanfiction is often highly derivative, creating new casts of characters, settings, and plots.

Due to the migration of fandom to social media, fans have the opportunity to communicate on discussion boards and blogs such as Tumblr and Discord, thus opening up fanfiction writers to more collaborative experiences (Gray et al. 2017). The evolution of fandom has required fans to continually redefine what it means to be a fan and to reflect on how their experiences are shaped by personal and interpersonal dimensions. While fandom is still most commonly described as "an investment of effect and identity into an object" it is also defined by the relationship and community fans establish with each other (Lothian 2018, p. 373).

Though fans have had the opportunity to write collaborative fanfiction for several decades since the introduction of the internet, its rise in popularity over the last 10 years may represent a cultural shift towards investment and emotional connection from commercialized media to the investment and creation of fanworks. In recent years, it has become more common for large groups of fans, with no prior connection to each other, to co-create fanfiction that introduces new lore, characters, and settings to the original media. Together, fans share their ideas and "poach" off of each other to create expansive worlds and stories that are open to be enjoyed and added to by the public, thus creating a large, interconnected narrative.

What I call, "inter-fan poaching" represents the process by which fans share and borrow narrative elements from one another in order to create a more unified story. In order for a story to be cohesive, fans must often bypass certain formalities and social norms in order to continue the story. As fans collaborate on large projects, they can sometimes contradict each other and create inconsistencies within this narrative. They create an identity of unity and collaboration through their usage of social media and how they negotiate narrative differences and contributions with one another.

These collaborative fictions then create a new hyperdiegetic space in which contribution to the narrative is based on the possibility that "it could happen" in the narrative (Hills 2002, p. 137). They break the binary of typical canon/fanon thinking in fanfiction and invite each other to enjoy the benefits of semi-canonical structures alongside the personal freedom to write their individual desires.

In this article, I discuss how these collaborative fanfictions introduce these new methods of inter-fan poaching and how hyperdiegetic narratives challenge ideas about who "owns" a text and what makes up its core canon. I argue that collaborative fanfiction forgoes the notion that there can only be a sole authority who owns and creates the rules of a story and instead highlights new methods of collective transformation. That is to say, collaborative fanfiction allows writers to make room for each other in a story. They can share ideas freely or they create space for each other to explore new thoughts and ideas—even if those ideas do not always make sense together.

My case study on collaborative fanfiction explores *Gravity Falls* (2012) Transcendence AU: a series of interconnected fanfictions, art, and videos written by different fans, ongoing since 2014, that explores a semiautonomous narrative that needs not always rely on the characters, setting, or plot in *Gravity Falls*. By analyzing this fandom, my objective is to demonstrate that fans are creating new hyperdiegetic narratives in their fanfiction, which allow a simultaneous space for both personal creative fandom and community collaboration.

## 2. Methodology and Case Study

*Gravity Falls* follows Dipper (Jason Ritter) and Mabel (Kristen Schaal) as they spend the summer at their Great Uncle Stan's (Alex Hirsch) Mystery Shack, a tourist trap based around the supernatural. After finding a hidden book called Journal 3, the twins investigate their new surroundings, uncovering a variety of dangers, including the demon Bill Cipher (Alex Hirsch) who seeks to destroy their universe and tear their family apart. The creation of Transcendence AU began during the show's second season, specifically starting from episode 24 "Sock Opera", which aired on 8 September 2014. In the episode, Mabel decides to put on a sock puppet rock opera to impress a local puppeteer, but her show goes astray when Dipper's drive to uncover journal secrets leads him to become possessed by Bill Cipher.

On 17 September 2014, Tumblr user zoey-chu made the initial post that sparked a new fandom, encouraging fans to contribute to an ongoing fan-made narrative:

> "Imagine in a canon-divergent AU, a huge, grand sort of finale where the Pines prevent an Armageddon-scale disaster. but some shit still goes wrong, and while they may have prevented the worst from happening, a huge event essentially unleashes a wave of supernatural phenomena . . . And during the Pines' "final battle" before all this happens, Bill pretty much dies . . . Dipper is also in pretty bad shape, vulnerable enough for Bill to attempt a last-ditch possession in order to save himself. But it doesn't work . . . not quite. A part of Bill ends up fusing with Dipper, but his mind isn't overtaken. It seems like Dipper lucked out and retains his mind, but his body isn't so lucky. He becomes a spirit much like Bill was, only able to reside in mindscape/underworld/whatever you want to call it. The result is bittersweet. Dipper can no longer return to the life he once had . . . Another curious side-effect is that tiny bits of Bill's personality and habits are picked up by Dipper."

Within months, what started as the musing of an idea became a collaborative project involving the creation of original plot lines and characters by hundreds of contributors and thousands of readers. While Transcendence AU began its life as "traditional" fanfiction that explored the world and characters by diverging from the original story and exploring the characters in new situations or settings, it soon became unrecognizable as *Gravity Falls*, forging references or explorations of the cast and setting and instead focusing on fan-made characters and places. Transcendence AU displays a clear divide between itself and the *Gravity Falls* fandom, with the fans having kept careful online records of their own history and narrative. The group is relatively small, with around 4000 followers on Tumblr with roughly 50 of those members active at a time. The community sees itself having broken off from *Gravity Falls* and now existing as an entirely separate fandom.

As a member of the Transcendence AU (TAU) community, I have developed close relationships with other fans, some of whom have volunteered as participants for this research. I am aware of my identity as a fan, producer, and scholar within TAU and the insight given by my personal investment and relationships in the fandom.

My relationship with Transcendence AU served as the inspiration for this research. Particularly, I was interested in how my community negotiated the narrative within the AU. I often noticed, through Transcendence AU's Tumblr blog and Discord server, that there were moments of negotiation between its members when it came to determining what the narrative should look like, where there were points of contention, and how the community should create a simultaneous space for unity and disagreement. That is to say, the majority of my analysis is my observations as both a participant and a scholar.

My personal relationship with the community allowed me some transparency in my research for this article and the larger thesis it is taken from. This open communication allowed me to communicate directly with members about using their social media posts in this article and if they would like their screen names to be used in the paper. The transparency also made my methods and ethics clear to the community, explaining what I was researching and what the examples I chose were demonstrating. It allowed me to treat the community respectfully and created a level of trust between us.

My personal insight into the community provided some first-hand experiences with how collaborative fanfiction operates. While I have written fanfiction for Transcendence AU and have been involved in the community since 2017, I avoid using examples of my own experiences in this article. I wish to provide examples that I had no hand in creating and to show the community from a purely observational stance.

In order to demonstrate the creation of Transcendence AU, account for its history, and learn about the nature of participants as both fans and producers, I employed an analysis of social media such as Tumblr and Discord. I specifically looked for instances in which I noticed fans acknowledging that they had borrowed ideas from each other, or instances in which there was debate or confusion about a canonical idea within the story. I followed online dialogues with the intention of writing this article in early 2019, and continuing up until June of 2022, to ensure that all of my examples were the best or most recent I could find. I primarily used the Tumblr blog (transcendence-au) and two Discord servers (transcendence_au, TAU 18+) to gather examples of fans communicating with each other.

Though, as previously stated, much of my understanding of this community comes from being a fan myself, because of my position as a member of the community, I was able to rely on my own memories when it came to selecting examples for this article. I used these texts to observe how fans interacted with each other and the AU's content to create collaborative stories by observing how they responded to blog posts, asked questions, and archived their history. While I sorted through hundreds of fanfictions, blog posts, and chat logs, in order to be concise, I chose only the most relevant and applicable examples for this article.

I specifically chose Transcendence AU for its transparency. The community puts great effort into tracking new fanfictions/ideas, as well as openly discussing these new contributions with each other through social media. This allowed for easy access to the

group's history and the ability to track how these collaborative fanfictions are formed. I also analyzed and reflected on current fanfiction hosted on Archive of Our Own, under the tag *Alternate Universe-Transcendence (Gravity Falls)*, to observe how fans reference each other directly in the summaries or notes, or indirectly through contextual references, repeated characters, and ideas.

### 3. Defining Collaborative Fanfiction

While research on collaborative fanfiction has been done before, it has predominantly focused on pairs of authors (or a small group of close friends) working to create a single work, rather than a collection of cohesive works. Thomas (2007) details the experiences of two teenage girls who co-write fanfiction together. The discussion raises important considerations, such as how the girls edit each other's sections "that do not cohere with the plot and insert[s] other lines to foreshadow what she knows will later become important to the narrative" (p. 144). Thomas also details the importance of technology and how the girls are able to communicate through Instant Messenger and use technology to write and edit the same document.

However, this case study does not fully represent the shift that has entered the fanfiction community. The two girls discussed by Thomas had met online and were friends prior to co-writing their fanfiction. They also write in a role-play format, which are "narrative threads that act as a hybrid of fan fiction and online gaming" (Howard 2017) that primarily represent character dialogue and actions and are transmitted back and forth between participants. While role play itself is a form of fanfiction, it does not represent the collaborative nature I discuss in this article.

Rather, I believe that collaborative fanfiction is defined by having several stories, written by different authors, combine into a larger narrative. These fans treat each other as if they are in a writer's room of a popular television show. They must do their best to honor the previous "episodes" written by other members and work together to create a sense of cohesion between stories. Collaborative fanfictions could be best described as a spiral of recursive fanfiction (or fanfiction about fanfiction). As fans respond to and build off of each other's writing, a new narrative is built from their ongoing and open communication.

However, these fans might not always be in direct communication with each other about the plot or end goals of a narrative. Instead, they may use the same characters or conflicts to show that their stories are various interpretations of the same base concept. For this reason, I do not include co-authored stories or role play in the definition of collaborative fanfiction because they do not contain these individual stories or "episodes".

Booth (2015) outlines an example within the Inspector Spacetime community, in which fans created a *Doctor Who* parody (originally sourced from the TV show *Community*) and then worked together to create a new set of narratives, characters, and worlds that they treated like a piece of popular mass media. He notes that following the appearance of Inspector Spacetime, fans immediately took to Twitter to discuss the fake show and later to Tumblr and TV tropes where they collaborated on creating the episodes, characters, and cast of the show, all of which used *Doctor Who* as the inspiration. Fans later created fanfiction about the Inspector's adventures, solidifying many of the shared ideas between fans into fully fleshed-out narratives.

These collaborative fanfictions can take multiple forms such as headcanons, fanon interpretations, ships, crossovers, parodies, and alternate universes. Alternate Universe fanfiction commonly relies on collaborative fanfiction, as fans add new lore and characters to a world and often borrow from each other to create their own expansive fictional universe. Transcendence AU began its life as "traditional" fanfiction that explored the world and characters by diverging from the original story and exploring the characters in new situations or settings; it soon became unrecognizable as *Gravity Falls*, forging references or explorations of the cast and setting and instead focusing on fan-made characters and places. Transcendence AU displays a clear divide between itself and the *Gravity Falls* fandom, with the fans having kept careful online records of their own history and narrative.



The community sees itself as having broken off from *Gravity Falls* and now existing as an entirely separate fandom.

While popular Alternate Universes, such as Coffee Shop AUs, can become vast collaborative efforts, they do not always push the boundaries of becoming their own fandoms in the way "shared universes" do. Shared universes are series of fiction stories written by multiple authors in the same alternate universe (Fanlore 2021), and often contain their own unique rules and other identifiable traits that set them apart from other Alternate Universes.

Collaborative storytelling can also be found in more extreme incidences, where the fandom's media object is created by its own audience. These fans create individual fanfictions that contribute to or expand upon previous entries, thus fleshing out the world building as they go. That is to say, efforts of collaborative fanfiction are pushing the boundaries of what a fandom can be and what happens when media objects are usurped (or wholly created) by the fans themselves.

While collaborative fanfiction can take many forms and does not need to focus on fans creating expansive and original universes for their fandom, the ability for fans to explore a fictional world encourages free thinking and creativity. While multiple fans can collaborate on a canon-driven fanfiction, the ability to break away from canon and build a universe mainly from scratch opens up more opportunities for more fans to fill the gaps or create new ideas.

## 4. Hyperdiegetic Narratives

Canon, within traditional media fandom, is a source of structure that provides the set rules or ideals that fans accept or reject (Gonzalez 2016). Canon describes the preserved memory and legitimacy of a culture, and in the case of fandom, it refers to the source material: the original book, film, television show, comic, etc., on which fanworks are based. Canon creates a contrast between what is "non-canonical", or untrue in the source material (usually meaning fanworks), and what is considered "canonical", or true (De Kosnik 2016, p. 104). The acceptance of a canon is considered respectful to media producers, and the ability to follow it is a demonstration of superior knowledge (Gonzalez 2016). However, this binary thinking between canon and non-canon does not always translate to collaborative writing projects.

Collaborative fanfictions instead expand upon Matt Hills's concept of hyperdiegesis, or "a vast and detailed narrative space, only a fraction of which nevertheless appears to operate according to principles of internal logic or extension" (Hills 2002, p. 137) by which fans touch upon the unexplored narratives within a canon. It allows for production practices such as discussion, speculation, and fanfiction within a media text's universe (Johnson 2017, p. 370). This imaginary space exists within all fandom and is vital to interactive and exploratory thinking with a text (Jones et al. 2018). This hyperdiegesis gives fans a sense of "reasonability" to focus around but also creates space for contradictory storytelling.

While created by the shared interpretations of many fans, a hyperdiegetic narrative acts in place of a canon. It is unstable, often contradicting itself due to the contrasting writing of a large number of its participants. In the same sense that the source material can present opportunities for contradiction or confusion, as Matt Hills describes with *Doctor Who* (Hills 2014), so too can fan-produced narratives. Hills writes: "Rather than allowing Doctor Who to function as a unified hyperdiegesis, or activating these sorts of fan readings, assorted multi-Doctor stories permit the fan market to buy into anniversary commemorations in a range of ways." (Hills 2014, p. 109) in order to explain how fans are able to balance concerns about timelines, actors' visible aging, and other continuity errors in episodes featuring multiple doctors.

This description of "could this happen" is in line with Hills's discussion of hyperdiegesis and the possibilities of that which is unseen in a narrative. Because a text cannot please everyone, Transcendence AU attempts to create a space in which all fans have opportunities to embrace the possibility and sense of cohesion in the narrative. A word was created

within the Transcendence AU fans' lexicon for this purpose—"squishy"—meaning that an event, character, or thing cannot be properly canonized within the cohesive narrative due to the number of overlapping ideas, such as dates within the narrative.

However, in order to be cohesive, there must be some indication of rules or themes. A post on the Tumblr blog, written by Mod R, reads, "Do not let the distinctions of "canon" and "fanon" stop you. The canon for this AU has always been nebulous and squishy. Aside from some core rules that shouldn't be broken, nothing is 'less canon' than anything else."

In Transcendence AU, it is expected that fans follow a very basic set of rules (Dipper Pines is the demon of this narrative, not Bill Cipher as in the source material; or that reincarnation is part of the story), and the rest is all true to canon, regardless of contradictions to previous works. This puts Transcendence AU, and fanfictions like it, in a position of being a narrative with a contradicting and changing canon.

Fans working around the idea of "believability" in their fanworks rather than truthfulness of the canon is a familiar concept to fandom. The freedom to ask "what if" and "why not" allows fans the ability to poach from each other, without being weighed down by the concerns of creating accidental contradictions. The hyperdiegetic narrative allows for exploration beyond the constraints of canon vs fanon thinking and allows fans to borrow, or "poach", from each other as they would from the source material in order to maintain an ongoing sense of community and legitimacy.

## 5. Inter-Fan Poaching

The ability to poach encourages a new form of fannish intertextuality where fanfiction and fanart can interact with, reference, and borrow from each other. Just as Jenkins suggests, fans can borrow "only what is pleasurable" from the media, so can they from each other. Texts do not exist in isolation from each other. The depth of knowledge and the complexity of interweaving fanfictions create an intertextual story in which fans must have some background knowledge about previous characters, plots, and creators in order to fully embrace the narrative.

Henry Jenkins defines fans as textual poachers who do not simply possess "borrowed remnants snatched from mass culture, but [rather] their own culture built from the semiotic raw materials the media provides" (Jenkins 2006, p. 49). Taken from Michel De Certeau's description of poaching as a raid on literary works to enjoy only pleasurable meanings and aspects, poaching is an ongoing struggle for ownership and control over the meaning of a text between fans and producers. There is no limitation that textual poaching can only be performed on a commercialized text but rather is an act that allows access to the means of cultural production: any and all culture is available for poaching.

It is typically considered plagiarism to borrow from another fan's works, such as plot details or original characters, without asking for direct permission (Fiesler 2008). The term "poaching" itself refers to stealing from the elite; therefore, fans should never steal from each other. However, collaborative communities forgo this idea in favor of sharing their ideas. They embrace a concept of inter-fan poaching: that fanworks can be poached in the same manner as with corporatized media. Paul Booth refers to this as "textual encroaching" (Booth 2013, p. 153), however, I argue that the word "encroach" suggests connotations of trespassing and violating consent. Rather, I propose the term inter-fan poaching to imply that the act is performed exclusively and willingly between groups of fans.

Fans in Transcendence AU explore continuations of each other's works, rewrite previously written scenes, or create original additions as they would for any other fandom. Just as Jenkins proposes, the act of poaching is "an impertinent raid on the literary preserve where fans take away only those things that are useful or pleasurable" (Jenkins 1992, p. 9). Fans do not have to poach every element of the narrative, just the parts they find interesting or think deserve to be further explored. No permission is needed to expand on an idea.

In terms of plot, one fan who goes by the username ToothPasteCanyon is noted as saying she "ripped off" another popular Transcendence AU fanfiction "Reverse, Rewind, Rewrite" by MaryPSue (Archive of Our Own) by borrowing the general plotline but flipping

the tone from being charming and heartfelt to angsty and violent. Her author's note on AO3 openly acknowledges that she has borrowed major concepts and ideas from another author, as well as the beta reader who helped her:

> "This work is based on the amazing Return, Rewind, Rewrite by MaryPSue. Go and read it here . . . Also a big thanks to StarlightSystem for helping me edit this story!".

In the Transcendence AU discord, ToothPasteCanyon also self-describes her fanfiction in the community Discord server as a rip-off, saying "I kinda rip off people's ideas [ . . . ] I'm standing on the shoulders of giants" and "I love ripping off [Reverse, Rewind, Rewrite]".

However, rather than be berated for her so-called "rip off," this was celebrated within the fandom and even popularized the idea of reusable plot lines that anyone can use without permission. Following in her path, several other fans began to "rip off" the same idea, borrowing the fanfictions' characters or general plot and adding their own original twists, thus pushing the narrative further.

Some fan-made creations are even considered synonymous with Transcendence AU; they are just as much a part of the narrative as the canonical characters are. For example, many fans referred to Henry, an original character who marries Mabel Pines (a canonical character from *Gravity Falls*) as an example of true "open use" within the community. He is an expected part of the story, as so many fans have written about or included his character within their own stories. This character was accepted into the canon simply because he was "first". His character was added by a fan relatively early on into the AU's creation, and thus became a staple within TAU with little contention.

Within Transcendence AU, original characters and ideas are readily accepted into the narrative so long as that character is not replacing or usurping another. Simply put, a rule of politeness is followed. It would be considered rude to destroy, kill, or alter another fan's original character, as it would also be rude to prohibit an original character from entering the story.

The ability to poach from each other allows fans the opportunity to create independent works that contribute towards a collaborative goal. Inter-fan poaching in these communities acts as a sign of respect, demonstrating that the creators' opinions and contributions have been appreciated and reified as part of the new text. Everything within the community is meant to be shared. It is a narrative that steals from itself.

## 6. Producing a New Canon

When fans create, there is often an ongoing "struggle for discursive dominance . . . over interpretation and evaluation" (Johnson 2017, p. 370) through which fans attempt to codify their beliefs about a text as the most important or prevalent "truth" in their community. Jenkins writes that fans may engage in heated debates surrounding interpretations of texts that all exist within a shared frame of reference about "what questions are worth asking and what moments provide acceptable evidence for these questions" (Jenkins 1992, p. 137). The most popular or approved interpretations or evidence are what lead fans to create what is traditionally called a fanon.

Derek Johnson refers to the process of competing through and comparing fanworks for the intention of creating the dominant narrative as "fantagonism" (Johnson 2017). This fantagonism can come at the cost of isolating or invalidating fans who do not belong to the majority of a fandom: white, middle-class, non-disabled Americans. Though fandom often intends to leave space for people of all cultures and identities to participate, the dominant interpretation of a text (even a fan-made text) can close off opportunities for critique, reexamination, or experimentation.

I do not mean to imply that all forms of fantagonism are meant to engender rifts between fans or stir up intentional controversy, rather the dominant interpretation often provides balance and structure to a story. In Transcendence AU, it is common for fans to communicate ideas with each other or ask questions about the "rules of their narrative". It is not uncommon for members to use the Discord server or the Transcendence AU Tumblr blog to ask questions about particular characters or plot events within the narrative. They

rely on dominant interpretations to create a sense of cohesion, so as to not drift so far away from what other traits the fans identify the AU with.

For example, user FreshMorningCoffee posted the question "Is there a specific year range that summoning demons is strictly prohibited?" in the Discord server. Other members of the community held an open discussion about their interpretations of the question at the answer:

> ToothPasteCanyon: I don't think we have specific eras when demons are banned! I imagine it's probably something always regulated but not always tightly cracked down on

> FreshMorningCoffee: I thought there were a few fics that implied it but wasn't quite sure

> RogueCHINCHILLA: I'd imagine a very large rise in regulation after California bit other than that its very up in the air

> ToothPasteCanyon: some people may have specific date ranges for sure!

> FreshMorningCoffee: Thank you! This really helps a lot

> Reynier: I do if it helps! According to my fic, in most of Europe summoning demons is banned from the early 2800s through the revolutions of 3148

> ToothPasteCanyon: !!!!!!!!! Oooo cool!

While some community members looked for previous fanfictions where an answer might be mentioned, thus following a dominant interpretation, others helped by suggesting ideas or theories based on what they have already read. While the answer was debated, it was not in a way that invited fans to disagree. The fans' belief in a "squishy" narrative takes the pressure off of creating a hard-set timeline and instead encourages fans to interpret the answer for themselves if they so choose.

Of course, some additions to the canon do not require debate or research. As previously stated, introducing original characters to the narrative is relatively easy, requiring only that a new fanfiction (or comic) be created to introduce that character. Since characters make up the heart of any fanfiction and also can be considered deeply important or personal to their creators, their integration into the universe is carried out with little debate.

### 7. Conclusions

Fanfiction practices have always relied on communal methods of creation. Brittany Kelley writes that "fanfiction communities more often depict an author who is never fully singular—always existing within complex, personally-engaged communities" (Kelley 2016, p. 53) in regard to the ways fans help each other through brainstorming and beta reading. However, this sense of author multiplicity is on the rise in current-day fandoms—being taken to extremes by fans who make their own media object and the fan works surrounding it. Collaborative fanfiction rejects the idea that "fans see themselves in highly individualistic terms" (Jenkins 1992, p. 88) and that authorship can only involve one creator with one linear story.

Fans working around the idea of "believability" in their fanworks rather than the truthfulness of the canon is a familiar concept to fandom (Hills 2002). Collaborative fanfictions take hyperdiegetic spaces to the extreme by being both fans and producers in those spaces. The freedom to ask "what if" and "why not" allows fans the ability to poach from each other, without being weighed down by the concerns of creating accidental contradictions.

Inter-fan poaching, in turn, works alongside hyperdiegetic narratives and celebrates the efforts and ideas that bring fans together. By passing new ideas back and forth without the social constraints of needing to ask permission to do so, they build rich worlds with interesting characters. While hyperdiegetic narratives make room for discussion and debate, inter-fan poaching provides a space for synergy that connects all story threads together.

By employing new practices of hyperdigetic narratives and inter-fan poaching, fans create new avenues for including each other in their story telling—be it through adopting their ideas into a fanfic or by making room to respectfully contradict each other.

**Funding:** This research received no external funding.

**Institutional Review Board Statement:** Not applicable.

**Informed Consent Statement:** Not applicable.

**Data Availability Statement:** Not applicable.

**Acknowledgments:** The author would like to thank Paul Booth and Samantha Close for their advice and guidance on the masters thesis this paper is derived from, and to the fans who inspired this work.

**Conflicts of Interest:** The author declares no conflict of interest.

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
