# Peer review of "Examining Collaborative Fanfiction: New Practices in Hyperdiegesis and Poaching"

_humanities, doi:10.3390/h11040087_

Round 1

Reviewer 1 Report

The article presents a new and interesting topic of the tendencies of collaborative fanfiction writing. It also opens the debate about the nature of collaborative fanfiction writing. However, there are several major points to reconstruct and rethink about this study:

  1. The objectives or research questions of the study are unclear. You have to clearly state what is the objective of the research
  2. There is no methods section. This is the crucial point of any research. You have to explain how you have made your bibliographic search and how many papers you found on your topic.
  3. The new knowledge produced in the paper is unclear. Featuring the results it is important to make clear  what is the new knowledge produced.

Author Response

I appreciate your concise feedback on this article. I hope that my revisions and further additions help to clarify and solidify the objective, as well as present my methodology. Please do not hesitate to express any further concerns you have about the research. Thank you for your time. 

Reviewer 2 Report

This article discusses the practices of collaborative fanfiction communities, in which writers work together to expand on fan-created universes instead of individually working on fanfiction based on traditional published canons. The article discusses how such communities build their own fan-created worlds, drawing on a shared notion of “believability.” They often embrace the instability and discontinuities of Matt Hill’s concept of hyperdiegesis. Collaborative fanfiction challenges traditional notions of fanfiction’s relationship to canon, as fans expand and “poach” material from other fan works, rather than from canon. The article examines examples primarily from the fandoms of Inspector Spacetime and Transcendence AU, but also mentions a few other fandoms.

Collaboratively built, fan-created worlds that have little or no link to published canons is a very intriguing area of fanfiction. This article makes a real contribution by focusing attention on these areas. The author makes the compelling point that this area has particular value in challenging traditional dichotomies between canon and fandom. If fanfiction is writing that is based on a media object, what happens when that media object is created (wholly or partially) by fans themselves?

However, the article does not seem ready for publication. Details about methodology and definitions of concepts are left unclear. Essential questions about the process of collaboration and decision-making in such fandoms are also left unanswered. I will expand on these issues below in numbered sections. Since the paper is based on a master’s thesis, I suspect there may be enough material to address these weaknesses in a timely fashion, but this also might not be the case.

  1. Lack of clarity in data and methodology

The article is not clear about what argument is being made about collaborative fanfiction, what communities are studied, and what methods are used. It would greatly aid the reader to know up front what the paper is doing and how the author comes to the conclusions that they do. The abstract and introduction lay out the topic (collaborative fanfiction) and what conclusions are made (that such fandoms “create a new hyperdiegetic space” and “break the binary of typical canon/fanon thinking,” but does not properly discuss how the research is designed to get to those conclusions. What examples of collaborative fanfiction communities are examined? What “data” does the author use to analyze these communities and come to these conclusions (examination of practices, textual artifacts, etc).

It would be helpful to clarify which example communities are studied, and how, early on. Lines 109-110 mentions that the focus of the analysis is on the examples listed in the previous paragraphs of Section 2. Bringing this up to the introduction, explaining why such communities were selected (likely because they are based on fan-created work instead of canon), and what “data” from the communities was examined to come to the research’s conclusions, would aid the reader. 

Hyperdiegesis is a fruitful theoretical angle to explain collaborative fanfiction, but it also explains how fans are able to generate so much fanfiction from traditional canon as well (as the author notes in Lines 133-134). The article argues that a “hyperdiegetic narrative acts in place of a canon” (Lines 137-138) in collaborative fanfiction, but how are such narratives different from published canons, which are also hyperdiegetic narratives? Perhaps the unifying narratives in collaborative fanfiction exhibit this “hyperdiegetic-ness” to a greater degree with their instability and contradictions? Clarity on this point would be helpful.

2. Lack of clarity of working definitions of concepts

The second main issue is a lack of clear working definitions for concepts that are critical to the article. It would be very helpful to more clearly define what the author means by “collaborative fanfiction”, the focus of the study. What is it and importantly, what is it not? The conclusion includes co-authoring and beta readership within collaborative fanfiction, yet the goal of section 2 is to contrast co-authoring with collaborative fanfiction. Can this be more carefully delimited? Perhaps a diagram showing overlap and disjunction would show how these terms are being conceptualized. I’m not totally sold on the distinction between collaborative fanfiction and co-authorship. Doesn’t roleplay often essentially involve interaction with texts that others have written (as their own character)? They also do not get to choose what other writers have said through other characters, but must accept it and work with it (Lines 68-69).

Transcendence AU is raised as a fascinating prototypical example of collaborative fanfiction. But what about popular “AUs” that are not specific flushed-out universes with fandoms but rather sets of tropes that are applied to many fandoms, such as Omegaverse or the “Soulmates” AU? This leads to a broader question about distinctions between collaborative fanfiction and “fanon”, which also has elements of widespread agreement between fans on non-canonical material. Perhaps there is a spectrum here with dimensions such as distance from canon (if there even is one) and individual creations vs collaborative creations? In any case, situating what exactly the paper is studying among these terms (those listed on Line 81, for example) would be very helpful. Also, in the case of making new fandoms from whole cloth, not from a published “canon” (such as Ships of the Northern Fleet as described on Lines 101-104), are there worlds that don’t have fandom qualities that also are collaboratively built? What are the fannish qualities here that separate this from simply a group of friends coming up with a fantasy story together?

3. Lack of explanation and detail on the processes of collaborative fanfiction

A third issue is a lack of explanation of the processes of collaborative fanfiction. The discussion of the practice of inter-fan poaching is insightful, but left me questioning how exactly collaborative fanfiction happens. Paired with hyperdiegesis, the concept of “believability” is useful here, but again more clarity is needed on the process: what is believable and what is not within these collaborations? How are the contours of the fan-created “canon” accepted and decisions made in these collaborative settings? With the example of Henry, spouse of Mabel Pines in Transcendence AU (Lines 212-213), how was this character proposed and why was it accepted over other possible characters? If anything could be submitted to Ships of the Northern Fleet, what fan stories are accepted by the community as central? It is very interesting that these collaborative fanfictions often embrace discontinuities in their “canon” (such as with Inspector Spacetime on line 168), but there must be some sort of process for who decided to create an episode of “The Three Inspectors” over other ideas. How is this “canon” produced and accepted? This crucially relates to the note about fan inclusion varying based on race, gender, social class, and disability (Lines 249-250).

Also, how are some communities alright with poaching whereas more traditional fanfiction is not (as described in Line 194)? Is this ever mentioned in fan materials that describe these communities, or how do fans implicitly understand “the rules” of one fan community versus another? Lines 158-161 mention that Booth lays out 3 tasks of collaborative fandom in the Inspector Spacetime fandom. How were these accomplished? Such detail would flush out the reader’s understanding of the practices of collaborative fanfiction. Or if this is out of scope for the article, a mention of why and a clearer definition of the scope of the article would address this.

These issues may be too much to address in one round of major revisions, but since there is a master’s thesis behind this article, there may be more material that could address these issues rather quickly.

Minor points:

  • Though it is certainly not critical to reference, work on fanfiction from the field of human-computer interaction may be helpful for thinking about the collaboration that happens around fanfiction. Two example articles are:
    • Campbell, J. A., Aragon, C., Davis, K., Evans, S., Evans, A., & Randall, D. P. (2016). Thousands of positive reviews: Distributed mentoring in online fan communities. Proceedings of the ACM Conference on Computer Supported Cooperative Work, CSCW, 27, 691–704. https://doi.org/10.1145/2818048.2819934
    • Dym, B., Brubaker, J. R., Fiesler, C., & Semaan, B. (2019). “Coming out okay”: Community narratives for LGBTQ identity recovery work. Proceedings of the ACM on Human-Computer Interaction, 3 (CSCW), 1–28. https://doi.org/10.1145/3359256
  • Lines 240-245: though the claims about Covid increasing collaborative fandom through online tools seem plausible, they also seem distracting from the main argument. A brief mention of a statistic for evidence of this would justify its inclusion, if possible.
  • This may be an issue that the journal formatters will handle, but you’ll want to look into how book names and TV show names, etc, should be formatted in this journal (italicized, etc), including in the references
  • Line 17: no apostrophe is needed after “others”
  • Line 27 “migration of fandom to social media”
  • Line 89: “though it soon became unrecognizable” (or some other grammatical change to connect to the previous clause)
  • Line 187: Michel De Certeau (no “a” in Michel)

Author Response

I appreciate your detailed feedback on this paper, especially the specific lines you are referring to. As a new scholar submitting for the first time, I found this to be particularly helpful advice. 

1. Methodology and Clarity in Data 
I hope that the methodology/case study I have added draws more attention to my methods in this article and how I gathered my information. You will also find that I chose to eliminate some of my case studies, Ships of the Northern Fleet and the SCP Foundation, and instead focus on Transcendence AU, of which I believe I have the most information. Of course, if it does not meet your standards or you still have questions do not hesitate to share them with me.  

2. Working Definitions 

I found your feedback here particularly insightful. I have reframed my explanation of these collaborative fanfiction communities by explaining my reasoning in more depth. Your comment about using Omegaverse or Soul Mate AUs was incredibly helpful, and I included more comparisons to popular AUs to explain my working definition.  I hope that I correctly interpreted your advice here. 

3. Defining the Collaboration Process 

Please let me know if any segment in the new section I added is unclear or does not answer your original questions. I briefly included a section connecting to Derek Johnson's definition of fantagonism, and how the negotiation process works for fans. However, I fear that it may veer off course from my initial argument (or perhaps not provide enough connection to the argument). Your insight here would also be much appreciated. 

Thank you so much for these minor revisions. Quickly correcting this errors is a major help. Tour suggestion to include some background information on human-computer interaction is fascinating, and I may follow through with that. However, I would like some more time to review the subject before deciding if I should include it in this article.  

Again, thank you for your time and precise, useful feedback. 

Reviewer 3 Report

Your paper is an highly enjoyable overview of a relatively new form of collaborative fanfiction writing in which authors are engaged in writing narratives that may be very distant from any canon material in traditional media but are instead part of a fan-built universe of ideas and narratives that all are free to build upon and borrow from. Your key contribution here is building on Jenkins' concept of textual poaching and proposing the concept of inter-fan poaching to describe how fanfiction writers consensually borrow ideas/characters/narratives from each other in the process of creating independent fanworks that contribute towards the collaborative community narrative.

Your writing style is clear, concise, and very considered. You use some particularly beautiful phrases and analogies throughout.

Your engagement with previous literature and concepts relevant to fanfiction studies was excellent, and I was very impressed with how you brought together a number of theories and concepts in your work. Your citations were all appropriate and showed excellent knowledge of the most up-to-date research in the field. Your work is well-argued (although see comments for improvement below) and your conclusions are appropriate and compelling.

In the attached PDF document, I have flagged some areas for improvement, and I will summarise them here:

Improvement 1: You need to revisit the idea of the individualistic nature of fanfiction presented in your introduction (see comments in the PDF).

Improvement 2: It is very late in the article that you actually present your key point. It is not until the section on Inter-fan Poaching that the reader understands what your contribution is and what your purpose is in writing the article. You need to foreground this idea much earlier, ideally in the introduction. This is my key criticism.

Improvement 3: It would be fantastic if you could give the readers a little more information about the Inspector Spacetime and Transcendence fandoms. I'm not familiar with either of these communities and I had to work quite hard to fill in the gaps on occasion. I was also left with some unanswered questions (see comments) about the nature of these fandoms. Similarly, there were a few parts in your paper (flagged in the attached) where it felt a little "data-lite" and an example or two would have really helped you bring your argument to life.

I truly hope that these are constructive points for you, and you don't feel disheartened by the feedback. This paper was a joy to read and you are a magnificent writer.

Author Response

I appreciate your precise feedback, especially the PDF inclusion. I found your questions and comments particularly useful. I hope I was successful in foregrounding my objective sooner within the article, but please let me know if it remains unclear.  

I also took your feedback to further expand my working definitions of individualistic fanfiction/ collaborative fanfiction by making several major changes and additions. I've added a methodology and narrowed down my case study to just one example: Transcendence AU. I hope that this provides more clarity and detail to my argument. I also hope that my specific examples of how these fans poach from each other and contribute to an ongoing canon provides more useful data. Of course, if you find any of this to be unclear or insufficient, do not hesitate to mention it. 

Again, thank you for your time. Your feedback was much appreciate. And thank you as well for the kind comments throughout. As a new scholar looking to publish in a journal for the first time, your little compliments went a long way. 

Round 2

Reviewer 1 Report

The paper improved immensely, however still major revisions have to be made. The objectives of the study are not clear enough yet. Please, indicate clear research questions or objectives. The methods section should be expanded: methods used to gather data should be included, the data gathered (all of the documents, screenshots, interviews, diaries, hours of observation, everything has to be quantified), the analysis has to be explained. Also, please, consider include "ethics" section and explain if you told the community that this study is being made, if the pseudonims are being used and so on.

In the results section, some quotes from the fandom that illustrate the points of the author are needed.

Conclusions section should be majorly rewritten. I feel that it is disconnected from the results. How poaching and Hyperdiegetic Narratives are included into the conclusions?

Author Response

Thank you for your time in reading and replying to this article, as well as for tolerating me as a new (and daunted) scholar. Your patience is appreciated. 

Point 1: Please, indicate clear research questions or objectives. 

Response 1: I have added an objective about my purpose behind the research, as well as strengthened my argument to explain how fans are rejecting the notion of a singular author or story. If you feel that this is still inadequate, please let me know and explain what I am missing or how it should be improved. 

Point 2: The methods section should be expanded: methods used to gather data should be included, the data gathered (all of the documents, screenshots, interviews, diaries, hours of observation, everything has to be quantified), the analysis has to be explained. Also, please, consider include "ethics" section and explain if you told the community that this study is being made, if the pseudonyms are being used and so on.

Response 2: I have added more commentary on my ethics and methods, though I fear that I might be unable to quantify as much as you may like. I hope to have made it more clear that my analysis for this paper is based more in personal experience and understanding of the community I am looking at. I did not pull from a specific sample of any social media posts, and instead navigated by memory for examples from conversations. Let me know how I can rephrase any of this to make my methods  clear. 

Point 3: In the results section, some quotes from the fandom that illustrate the points of the author are needed. 

Response 3: As stated above, I did provide some quotations that help illustrate my point. I hope they are cohesive within the argument, but I leave it to your judgement if I need more explanation (or even more examples) to make things clear. 

Point 4: Conclusions section should be majorly rewritten. I feel that it is disconnected from the results. How poaching and Hyperdiegetic Narratives are included into the conclusions? 

Response 4: I agree with your statement that my conclusion was disconnected. I worked to rewrite and include more of my opening statements in the article at the end. 

Round 3

Reviewer 1 Report

The paper improved drastically. Especially the conclusions are very well discussed. I also liked the extended methodology chapter where the authors relate their relationship with the community.

To make the study more vigorous, I would suggest to include more details into the methodology, data gathering section, e.g. what were the channels you followed, for what amount of time, how many texts did you gather (or how many words) etc.

Author Response

Thank you so much for your feedback. I've added more details into the methodology about when I started looking at the fanbase through a more scholarly lens and exactly what online serves I used. I also sought to briefly mention that I included the best/most recent examples of fan communications for this article in order to be concise. 
I also read through for any remaining grammar or spelling errors at your suggestion. 

Again, thank you for your time.